# No Significant Beneficial Effects of Intravenous N-Acetylcysteine on Patient Outcome in Non-Paracetamol Acute Liver Failure: A Meta-Analysis of Randomized Controlled Trials

**DOI:** 10.3390/biomedicines12071462

**Published:** 2024-07-01

**Authors:** Carmen Orban, Mihaela Agapie, Angelica Bratu, Mugurel Jafal, Mădălina Duțu, Mihai Popescu

**Affiliations:** 1Department of Anesthesia and Intensive Care, “Carol Davila” University of Medicine and Pharmacy, 37 Dionisie Lupu Street, 020021 Bucharest, Romania; carmen.orban@umfcd.ro (C.O.); jafalmugurel@yahoo.com (M.J.); madalina.dutu@umfcd.ro (M.D.); mihai.popescu@umfcd.ro (M.P.); 2Department of Anesthesia and Intensive Care, Bucharest University Emergency Hospital, 169 Independentei Street, 050098 Bucharest, Romania; angibratu@yahoo.com; 3Department of Anesthesiology and Intensive Care, “Dr. Carol Davila” University Emergency Central Military Hospital, 134 Calea Plevnei, 010242 Bucharest, Romania

**Keywords:** N-acetylcysteine, acute liver failure, drug-induced liver failure, survival

## Abstract

Acute liver failure is a life-threatening organ dysfunction with systemic organ involvement and is associated with significant mortality and morbidity unless specific management is undertaken. This meta-analysis aimed to assess the effects of intravenous N-acetylcysteine (NAC) on mortality and the length of hospital stay in patients with non-acetaminophen acute liver failure. Two hundred sixty-six studies from four databases were screened, and four randomized control trials were included in the final analysis. Our results could not demonstrate increased overall survival (OR 0.70, 95% CI [0.34, 1.44], *p* = 0.33) or transplant-free survival (OR 0.90, 95% CI [0.25, 3.28], *p* = 0.87) in patients treated with intravenous NAC. We observed an increased overall survival in adult patients treated with NAC (OR 0.59, 95% CI [0.35, 0.99], *p* = 0.05) compared to pediatric patients, but whether this is attributed to the age group or higher intravenous dose administered remains unclear. We did not observe a decreased length of stay in NAC-treated patients (OR −5.70, 95% CI [−12.44, 1.05], *p* = 0.10). In conclusion, our meta-analysis could not demonstrate any significant benefits on overall and transplant-free patient survival in non-acetaminophen ALF. Future research should also focus on specific etiologies of ALF that may benefit most from the use of NAC.

## 1. Introduction

Acute liver failure (ALF) represents an acute hepatic dysfunction, most often involving multi-system organ failures, and is associated with high mortality and morbidity unless rapid diagnosis is established and intensive care measures are undertaken urgently. The classic triad of ALF is represented by hepatic encephalopathy, jaundice, and coagulopathy [1], but depending on underlying etiology, patients may present varying degrees of liver or multiple system organ failure that are hard to differentiate from other causes, including sepsis and shock [2]. The management of such patients includes a combination of standard medical care according to current guidelines [3], extracorporeal liver support [4,5], and liver transplantation as the only definitive treatment in severe cases [6]. However, there are still some debatable issues regarding the appropriate management of ALF patients in terms of the exact indications and timing of liver transplantation, as well as the most appropriate medical management, as the standard of care has significantly changed in recent years [7]. Nevertheless, patient outcomes have significantly improved over the last two decades, with a decrease in mortality, more patients undergoing spontaneous remission, and fewer patients requiring emergency liver transplantation [8,9]. These trends have made well-known researchers consider that ALF has the potential to become a curable disease in the next few years [10].

The etiology of ALF has varied widely across the years and among countries, with paracetamol overdose being the most frequent cause in Western countries and viral hepatitis still having a high incidence in developing countries [11]. In recent years, there has been a significant increase in non-acetaminophen drug-induced liver injury (DILI), which represents a worldwide problem, as it has been associated with a worse outcome and decreased survival [12,13]. However, a significant number of patients with ALF are still classified as having an indeterminate cause, and specific etiology-oriented tests should be carried out for a better diagnosis and to ensure specific, etiology-based treatment regimens [14].

N-acetylcysteine (NAC) has long been used as a specific antidote in ALF due to paracetamol overdose. In this setting, it works by restoring glutathione stores and thus re-establishing the metabolism of N-acetyl-p-benzoquinone imine, a highly toxic by-product of acetaminophen metabolism, to inactive compounds. Many studies have focused on the potential use of NAC in other, non-acetaminophen forms of ALF. The pharmacological rationale for the use of NAC in these circumstances is based on NAC’s effects on either rebalancing the systemic inflammatory response or the ability to reduce oxidative stress, both key factors in the pathophysiology of ALF [15,16]. Based on animal studies that demonstrated the anti-inflammatory and antioxidant effects of NAC, subsequent trials were designed to test the ability of NAC to improve the outcome of patients with non-acetaminophen ALF [17,18]. However, we have observed a huge heterogeneity among studies in terms of the exact timing of therapy initiation, the route of administration (oral versus intravenous), and the administered dose. In the last decade, different meta-analyses tried to summarize current evidence, but they were unable to distinguish between studies involving patients with different severities of liver dysfunction or the administered dose and route. Hence, we aimed to perform a meta-analysis looking at the potential benefits of intravenous NAC in patients with non-acetaminophen ALF defined by well-established international criteria.

## 2. Materials and Methods

Our study was performed in accordance with PRISMA recommendations and was registered on PROSPERO—International prospective register of systematic reviews under the ID CRD42024533188. The aim of this study was to perform an updated meta-analysis to assess the benefits of intravenous NAC in the management of non-acetaminophen ALF. ALF is defined according to the criteria set by international guidelines [19]: a sudden loss of liver function secondary to an acute, catastrophic liver injury, manifested clinically by hepatic encephalopathy and jaundice and paraclinically by coagulopathy (defined as an international normalized ratio ≥1.5) and hyperbilirubinemia in a patient without a known history of liver disease.

*Outcome of meta-analysis*. The primary outcome of this meta-analysis was to assess the effects of intravenous NAC on decreasing mortality in patients with non-acetaminophen ALF. Secondary outcomes were the assessment of intravenous NAC on (1) transplant-free survival and (2) the length of hospital stay.

*Study eligibility criteria*. In the current meta-analysis, the following criteria were used for the inclusion of clinical trials. All randomized, controlled trials evaluated the impact of intravenous NAC on patient outcomes in non-acetaminophen ALF. To be included, trials should have recruited patients who were randomized to receive either intravenous NAC, standard of care, or placebo. The following were considered as exclusion criteria: use of oral NAC preparations; studies including patients with acute-on-chronic liver failure or other forms of liver failure not fulfilling the above-mentioned definition; and the etiology of ALF being post-hepatectomy, sepsis, or ischemic hepatitis. If ambiguity in the study methodology was identified, the corresponding author of the trial was contacted for additional information. If an identified study was proven to be an abstract from a recognized international congress, we tried to contact the authors for further information regarding the publication status of their article. A PICO frame to facilitate the understanding of inclusion and exclusion criteria can be found in Table 1.

*Search strategy*. The search was conducted by two reviewers in the following databases: PubMed, EMBASE, Scopus, the Cochrane Library, and Web of Science from the establishment of the database to January 2024. Language limitations were set to English only. The search terms included intravenous N-acetylcysteine, non-acetaminophen, non-paracetamol, acute liver failure, outcome, survival, transplant-free survival, and the length of hospital stay. The reference lists of identified publications that fit within the scope of our meta-analysis were also investigated to identify other qualified trials not found in the initial database search.

*Inclusion of studies*. After applying the search terms in all the databases, the identified studies were screened by two reviewers by reading the title and abstract alone. After this initial step, a second round of screening was performed to evaluate the full-text versions of all potentially eligible randomized trials in accordance with the inclusion criteria. Any disagreements were discussed alongside a third reviewer. All duplicate studies were eliminated. Preferred Reporting Items for Systematic Reviews and Meta-Analysis (PRISMA) statement guidelines were followed for conducting and reporting meta-analyses.

*Assessment of risk of bias*. To assess the risk of bias for the included publications, two reviewers independently used the tool provided by the Cochrane Collaboration (London, United Kingdom) [20]. The final rating of quality for each study was classified as high, moderate, or low. Any discrepancies in risk of bias were settled by the third reviewer.

*Data extraction*. Data were extracted in duplicate in a standardized data extraction form by two independent reviewers and compared at the end of the process. Any discrepancies in data extraction were discussed and settled by a third reviewer. The data were primarily extracted from tables found in the included studies. Data extraction included the following information: basic information (author name, year of publication, number of cases, age, etiology of ALF, dose, and duration of intravenous NAC), primary outcome (patients’ survival), and secondary outcomes (transplant-free survival, the length of hospital stay).

*Statistical analysis*. The statistical analysis was performed using the Rev-Man online tool (Cochrane Collaboration, Oxford, UK) available at https://revman.cochrane.org/, accessed on 20 April 2024. Results are expressed as odds ratio (OR) for dichotomous data or weighted mean differences for continuous data, both with 95% confidence intervals (CIs). Statistical heterogeneity was assessed by visually examining the forest plots and quantified using the I^2^ statistic. An I^2^ > 50% was considered to indicate substantial heterogeneity. We conducted a random-effect meta-analysis when there was significant heterogeneity; otherwise, we used the fixed-effect model. For effect sizes, the odds ratio (OR) for dichotomous outcomes and standardized mean difference (SMD) for continuous variables were calculated using a random-effect model in cases of significant heterogeneity between estimates. A *p*-value >0.05 was considered to reject the null hypothesis that the studies were heterogeneous.

## 3. Results

The initial search identified 364 articles; after duplicate removal and title and abstract screening, 30 publications were considered for full assessment. Of these, 22 were excluded because of various reasons, and one study could not be retrieved even after repeated requests were sent to the authors. Data were extracted from four studies [21,22,23,24] (two including pediatric patients and two including adult patients) that fulfilled the inclusion criteria. A schematic diagram for study selection criteria is presented in Figure 1. All studies were published as full articles. Two studies used a high decremental dose of NAC for 72 h, while two studies used the same low-dose continuous infusion regimen.

### 3.1. Studies and Patients’ Characteristics

Of the four studies, two were multicenter, of which one was conducted in the United States alone and one was conducted in the United States and the United Kingdom. The other two studies were single-center, one from India and one from Pakistan. The Cochrane risk-of-bias tool was used to assess the quality of the included RCTs and their associated risk of bias (Figure 2).

A total of 469 patients were included in the meta-analysis, with 228 in the interventional group and 241 in the control group. Of these, 46.0% (*n* = 216) were pediatric patients, and the rest were adults. The main etiology was acute viral hepatitis (25.1% of patients), followed by drug-induced liver injury (14.9% of patients), and 41.1% had indeterminate etiology. A summary of included studies is presented in Table 2.

### 3.2. Overall Mortality Assessment

Concerning overall mortality assessment, the I^2^ was 65%, demonstrating significant heterogeneity, and hence, we performed a random effects analysis. No significant difference was observed between the NAC group (28.8%) compared to the control group (33.75%) in terms of mortality (OR 0.70, 95% CI [0.34, 1.44], *p* = 0.33)—Figure 3.

We subsequently performed an analysis looking at the different patient populations. In the two studies conducted on adult patients, 253 patients were included: 121 patients in the NAC group and 132 patients in the control group. We observed a significant decrease in mortality associated with the use of NAC in this patient population (OR 0.59, 95% CI [0.35, 0.99], *p* = 0.05). There was no difference in mortality between NAC and control groups in the pediatric population (OR 0.87, 95% CI [0.20 3.85], *p* = 0.86). However, the adult population studies used a high-dose decremental NAC regimen compared to pediatric studies, which used a low-dose constant continuous infusion.

### 3.3. Transplant-Free Survival

Two studies assessed transplant-free survival: one pediatric and one adult population study involving a total of 173 patients in the NAC group and 184 in the control group. The I^2^ of 89% demonstrated significant heterogeneity, and we performed a random effects analysis. There was no difference between the NAC group and control group in terms of transplant-free survival (OR 0.90, 95% CI [0.25, 3.28], *p* = 0.87)—Figure 4.

### 3.4. Length of Hospital Stay

Although all studies reported on the effects of intravenous NAC on the length of hospital stay, data could only be collected from two studies, which included a small number of 56 patients in the NAC group and 56 in the control group. Although both studies reported a significant decrease in the length of hospital stay, pooled data were not able to demonstrate a statistically significant decrease in hospitalization (OR −5.70, 95% CI [−12.44, 1.05], *p* = 0.10)—Figure 5.

## 4. Discussion

NAC has widely been used in different types of liver failure to augment spontaneous remission of liver dysfunction. The pharmacological rationale for this is based on the effects of NAC as an anti-inflammatory and an antioxidant agent. In paracetamol-induced liver failure, it has been successfully used as a specific antidote to decrease the quantity of a highly active intermediate metabolite (N-acetyl-p-benzoquinone imine) by restoring glutathione that normally metabolizes it [25]. Nevertheless, both experimental and clinical studies have demonstrated that its therapeutic effects are well beyond re-establishing the metabolic pathway for paracetamol. Animal studies using resonance spectroscopy have demonstrated that NAC is effective in preventing liver injury, accelerates the recovery of both adenosine triphosphate and glutathione, and increases substrate flux through the Krebs cycle [26], thus assuring proper function of both lipid peroxidation and antioxidant enzyme systems [27]. These molecular mechanisms translate clinically into a reduction of the inflammatory cascade that represents the key pathophysiological mechanism behind organ dysfunction in ALF [28,29]. However, the problem at hand is whether, based on the significant experience gained from the paracetamol-induced ALF studies, “*one size fits all*” and NAC can be used safely in other etiological causes of ALF.

To date, there are many published case reports and case series on the benefits of NAC in non-APAP ALF, but most of these studies have major drawbacks: (1) there is a major heterogeneity of ALF definition across studies ranging from hepatocytolysis alone to patients fulfilling AASLD criteria; (2) the dose and route of administration varies widely; (3) one size does not fit all, and most of the major studies include different etiologies of ALF; (4) mortality outcomes differ in terms of duration (ranging from 28 days to 1 year), and moreover, they may be attributed to different factors not related to ALF itself (e.g., septic shock secondary to immunosuppression, etc.); and (5) the standard of care varies widely among centers and across time and has changed significantly in the last two decades, and this may have a significant impact on patient outcome.

Intensive care management of non-acetaminophen ALF patients remains a challenge due to the rapid progression of the disease, difficulties in determining the underlying etiology, the low number of expert centers, and the limited number of organs available for transplantation [30]. Thus, aggressive, early medical care remains a key aspect in assuring spontaneous remission and increased survival. Many studies and subsequent meta-analyses focused on the use of NAC as a non-specific, general measure in the management of ALF patients. However, in our opinion, these previous studies did not consider several important issues: First, they did not take into consideration either the route of administration (orally versus intravenously) or the administered dose, with dose regimens ranging from 600 mg/day up to 150 mg/kg/day [31,32]. Most importantly, many of the published meta-analyses did not take into consideration the severity of liver dysfunction. Thus, they included studies containing patients with different levels of severity ranging from mild hepatocytolysis or cholestasis to patients fulfilling current ALF criteria.

In their meta-analysis, Chughlay et al. [31] managed to include only one randomized control trial published by Lee et al. [21] in 2009 and could not find a significant difference in the overall survival, although they observed that NAC compared with placebo significantly improved transplant-free survival. However, their results were based on only this one trial and should be interpreted with caution. In a second recently published meta-analysis, Amjad et al. [32] included five prospective studies: three observational and two randomized double-blinded trials. Of the five trials, three used an intravenous NAC regimen, one used a mixed intravenous and oral route, and one used an oral route-only regimen. One of the included trials was that published by Darweesh et al. [33], who defined ALF by the presence of jaundice and coagulopathy alone, regardless of hepatic encephalopathy. Moreover, as most patients had no hepatic encephalopathy (71% in the treatment group and 64% in the control group), the majority of patients had acute viral hepatitis that, in general, has been associated with a more favorable outcome [34], and no severity scores were reported. In our opinion, it is difficult to assess whether the patients included in the study by Darweesh et al. truly fulfill the current criteria for ALF. Nevertheless, the results of this meta-analysis showed that NAC treatment significantly improved transplant-free survival, whereas the overall survival remained unchanged, and it also decreased the length of hospital stay by approximately two days.

Shrestha et al. [35] conducted one of the largest meta-analyses of 11 studies, including 1117 patients, of which 565 received NAC. However, the authors included both randomized and non-randomized trials, as well as both observational and propensity-matched studies. Also, there was significant heterogeneity in the severity of liver disease between the studies, ranging from patients with mild presentations having hepatocytolysis and mild cholestasis to patients requiring intensive care unit management and extracorporeal support. However, their results showed a significant decrease of 53% in mortality compared to the standard of care, as well as a 6.5-day reduction in the length of hospital stay. The authors also noted that more than half of the patients had an improvement in the hepatic encephalopathy grade. They reported an increased incidence of nausea and vomiting and a higher need for mechanical ventilation as the most important side effects in the treatment group. Although nausea and vomiting have been reported as a side-effect in different clinical scenarios in which NAC was used as a treatment option [36,37], the increased need for mechanical ventilation may be subject to debate, as most ALF patients require mechanical ventilation secondary to neurological dysfunction. In addition, the authors reported a significant decrease in the severity of hepatic encephalopathy, and hence, there should have been fewer patients requiring mechanical ventilation. As the reason for this may be an increased incidence of respiratory failure and acute respiratory distress syndrome associated with NAC treatment, further research is needed to completely understand these observations.

To address what we consider to be shortcomings of previously published reviews, we conducted a meta-analysis of high-quality randomized control trials in patients with defined ALF criteria who received intravenous-only NAC therapy as part of their medical management.

After database interrogation, 266 abstracts were screened for eligibility, and 30 articles were reviewed in full length to be considered for inclusion. However, we could not retrieve one article even after we tried repeatedly to contact the corresponding author. Of the 29 screened articles, the majority were excluded because patient inclusion criteria were not fulfilled in accordance with the ALF definition or the route of administration was both oral and intravenous. One additional article [38] was excluded because it reported results based on the same cohort of patients, and thus, four randomized control trials were included in the final analysis.

Our results failed to demonstrate any benefits of intravenous-administered NAC on overall mortality. This result was mainly attributed to the study by Squires et al. [22]. In their study on a pediatric population with ALF, the authors demonstrated no significant improvement in survival in patients treated with NAC but a trend, although also non-significant, toward a lower 1-year survival. We further performed an analysis of the two studies involving adult patients [21,24] and demonstrated a significant decrease in mortality associated with the use of NAC. However, a significant factor should be considered when analyzing our results: pediatric patients received a much smaller dose based on body weight compared to adult patients, and thus, this result may be due to dosing considerations and not to age-related factors. Also, there was a significant difference in the etiology of liver disease among pediatric and adult patients, so we consider that the observed difference between pediatric and adult patients may be multi-factorial and should be investigated in future studies. Another significant limitation of our analysis is represented by the different time intervals for survival reporting, with some studies reporting on one-month survival and another on 1-year survival. Thus, future studies should focus on early outcomes, such as one-month survival, as, in most cases, the natural history of ALF is short and NAC is used in acute management. In addition, longer mortality reporting should be taken into account only when considering other long-term treatment options, such as liver transplantation.

The second endpoint of our meta-analysis was to assess transplant-free survival. Only two studies reported on this outcome, and we found no significant change in survival. Interestingly, the study by Squires et al. [22], which reported no change in overall survival, reported a decrease in transplant-free survival. Moreover, one of the two studies included a report on the adult population, and the other one was on pediatric patients. However, although both studies were conducted in the United States, none of them presented the criteria for liver transplantation in the methods section, and therefore, we cannot draw any definitive conclusion on the exact impact of NAC on transplant-free survival, as we cannot assess whether the same objective criteria were used in all patients.

The last point we analyzed was the effect of NAC on the length of hospital stay, with only two studies and a small number of patients included. Our results did not demonstrate a decrease in the length of hospital stay associated with the use of NAC. However, some points should be made. The two studies were conducted in different regions and healthcare systems, and hence, we do not know if the same discharge criteria were applied. Moreover, as the discharge criteria were not reported in the methods section, we had no information on whether the decision to discharge was based on objective criteria or the attending physician’s decision, a subject of significant bias.

As previously mentioned, different meta-analyses were published, reaching different results based on the studies’ inclusion criteria. Amjad et al. [32] included ALF patients who received NAC regardless of the route of administration and observed decreases in transplant-free mortality and the length of hospital stay but not in overall mortality. In another meta-analysis, Jawaid et al. [39] analyzed three prospective trials on ALF adult patients and concluded that the use of NAC was associated with an increase in transplant-free survival with no significant increase in the incidence of major adverse events. The difference in observed results reported by our analysis and others [21,32,35,39,40,41,42] may be due to the vast heterogeneity in patients’ population and dose regimens of NAC.

The route of administration and dose regimen for NAC has been the topic of intensive debate. In older studies, NAC was mainly used in its oral form; however, subsequent trials preferred to use intravenous preparations because of the fear of ineffectively achieving a therapeutic plasma concentration after oral administration because of impaired absorption, delayed gastric emptying, and intestinal failure seen in patients with ALF [43]. This may be the reason why these first studies failed to observe any benefit of NAC administration in non-paracetamol ALF [44]. Because of this, most guidelines and international consensus statements argue for the use of intravenous NAC formulations [3,45] but fail to recommend a specific dose aside for paracetamol overdose. Nevertheless, as most studies yielding positive used a similar intravenous dose of 150 mg/kg over 1 h, followed by either 100 mg/kg doses every 6 h over a total of 72 h [46] or 12.5 mg/kg/hour for 4 h, then 6.25 mg/kg for 67 h, this regimen may be used until future research is available. However, this is not without risk. Although most studies demonstrated only mild side effects, as previously mentioned, a recently published experimental study showed that the use of high-dose NAC in both normal and ALF mice was associated with decreased levels of glutathione, increased levels of pro-inflammatory cytokines, and hepatic microvesicular steatosis, all of which were probably responsible for the increased number of deaths dose-dependently associated with the use of NAC [47].

One of the major limitations of our meta-analysis and the ones published before that we did not previously mention is that analyzed studies did not consider the underlying etiology of ALF. In this case, all studies included ALF patients of different etiologies, and pooled results were reported. This approach may include significant errors in the processed results as one size does not fit all, and NAC may not have a beneficial effect in all etiologies. A second important limitation is represented by the heterogeneity across time and geographical regions of included studies that may be associated with a significant bias in terms of the standard of care of ALF patients that has changed significantly in the last two decades. Other limitations are represented by the non-standardized criteria for liver transplantation when reporting transplant-free survival and the difference in patient age ranging from pediatric population to older adults.

## 5. Conclusions

Based on currently published data, our meta-analysis of randomized control trials could not demonstrate any significant benefits on overall and transplant-free patient survival in non-acetaminophen ALF. Although we observed a potential benefit for improved overall survival in the adult population receiving NAC, it is unclear whether this can be attributed to patient age or to the dose itself. Also, we could not demonstrate a significant decrease in the length of hospital stay that may be attributed to different discharge criteria used across studies. Our study highlights that, to date, there is no sufficient evidence to support the routine use of NAC in non-APAP ALF. To fully evaluate the effects of NAC on both overall and transplant-free survival, as well as possible dosing regimens, more well-designed studies are needed.

## Figures and Tables

**Figure 1 biomedicines-12-01462-f001:**
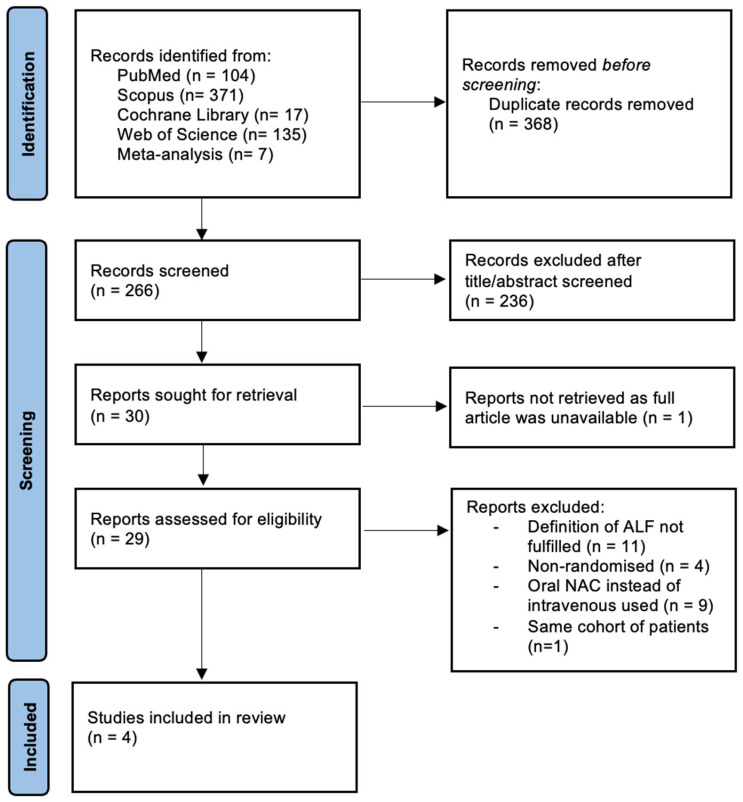
PRISMA flowchart of study selection based on the inclusion and exclusion criteria. Legend: ALF—acute liver failure; NAC—N-acetylcysteine.

**Figure 2 biomedicines-12-01462-f002:**
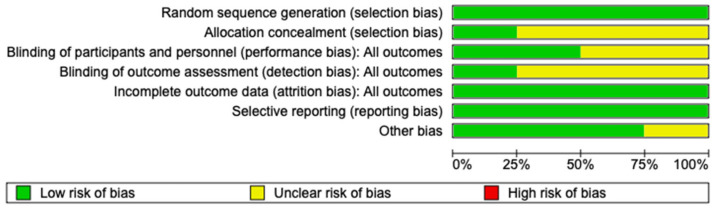
Summary of risk of bias.

**Figure 3 biomedicines-12-01462-f003:**
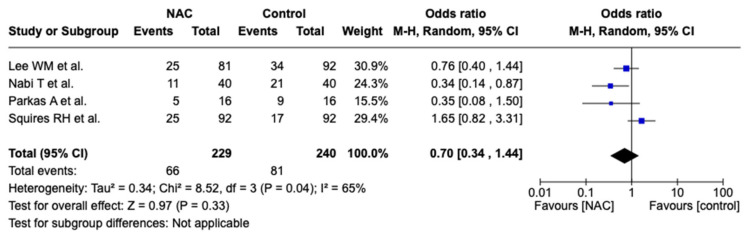
Forest plot for overall mortality. Blue squares represent odds ratio alongside 95% confidence intervals and black rhombus represents total odds ratio and 95% confidence intervals.

**Figure 4 biomedicines-12-01462-f004:**
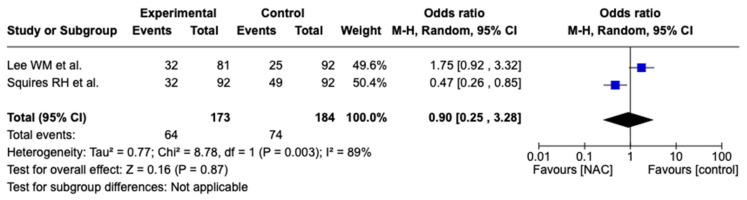
Forest plot for transplant-free survival. Blue squares represent odds ratio alongside 95% confidence intervals and black rhombus represents total odds ratio and 95% confidence intervals.

**Figure 5 biomedicines-12-01462-f005:**
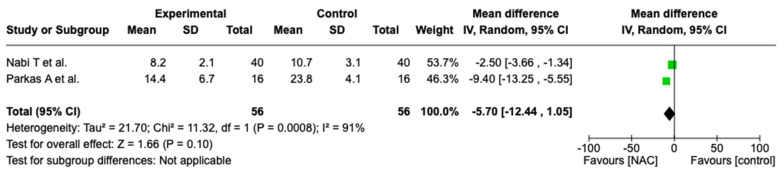
Forest plot for length of hospital stay. Green squares represent mean differences alongside 95% confidence intervals and black rhombus represents total mean difference and 95% confidence intervals.

**Table 1 biomedicines-12-01462-t001:** PICO frame for study design.

PICO Component	Inclusion Criteria	Exclusion Criteria
Population	- Patients diagnosed with non-acetaminophen ALF according to AASLD definition [19] regardless of etiology	- Other forms of liver dysfunction not fulfilling the definition of ALF (e.g., acute-on-chronic liver failure, acute hepatitis) - Postoperative liver failure - Liver failure secondary to shock
Intervention	- Intravenous NAC, regardless of dose	- Oral NAC
Comparison	- Placebo or standard of care	- No exclusion criteria
Outcome	- Mortality in non-acetaminophen ALF - Transplant-free survival - Length of hospital stay	- Studies not reporting on at least one of the outcome criteria

Legend: ALF—acute liver failure; AASLD—American Association of the Study of Liver Disease; NAC—N-acetylcysteine.

**Table 2 biomedicines-12-01462-t002:** Characteristics of included studies.

Publication	No. of Patients	Mean Age	Etiology	Dose of NAC	Outcome Reported	Survival
Lee WM et al., 2009 [21]	173 patients: 81 NAC group vs. 92 controls	40.5 y in NAC group and 42 y in control group	- DILI (*n* = 45) - indet (*n* = 41) - HBV (*n* = 37) - AIH (*n* = 26)	loading dose of 150 mg/kg/h over 1 h, followed by 12.5 mg/kg/hour for 4 h, then 6.25 mg/kg for 67 h.	- increased transplant-free survival - lower LT rate	70% in NAC group vs. 66% in control group
Squires RH et al., 2013 [22]	184 patients: 92 in NAC group vs. 92 controls	3.7 y in NAC group vs. 4.5 y in control group	- indet (*n* = 109) - AIH (*n* = 19) - metabolic (*n* = 18) - infection (*n* = 15) - other (*n* = 23)	150 mg/kg/day for up to 7 days	- no difference in 1-year survival, - lower 1-year transplant-free survival	73% in NAC group vs. 82% in control group
Parkas A et al., 2016 [23]	32 patients: 15 in NAC group vs. 16 controls	7.5 y in NAC group vs. 7.6 in control group	- HAV (*n* = 23) - non A-E hepatites (*n* = 5) HBV (*n* = 3) - HAV-HEV co-infection (*n* = 1)	100 mg/kg/day until normalization of the INR or death	- non-significant higher survival in NAC group - shorter hospital LoS	69% in NAC group vs. 44% in control group
Nabi T et al., 2017 [24]	80 patients: 40 in NAC group vs. 40 controls	30 y in NAC group vs. 38 y in control group	- viral hepatitis (*n* = 49) - DILI (*n* = 25) - indet (*n* = 43) - other (*n* = 6)	150 mg/kg for 1 h, then 12.5 mg/kg/h for 4 h and 6.25 mg/kg/h for 67 h	- better survival rate - shorter hospital LoS	72% in NAC group vs. 47% in control group

Legend: NAC—N-acetylcysteine; y—years; DILI—drug-induced liver injury; indet—indeterminate etiology; HBV—hepatitis B virus; AIH—autoimmune hepatitis; HAV—hepatitis A virus; HEV—hepatitis e virus; INR—international normalized ratio; LoS—length of stay.

## Data Availability

The data presented in this study are available on request from the corresponding author.

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
