# Peer review of "No Significant Beneficial Effects of Intravenous N-Acetylcysteine on Patient Outcome in Non-Paracetamol Acute Liver Failure: A Meta-Analysis of Randomized Controlled Trials"

_biomedicines, 2024, doi:10.3390/biomedicines12071462_

Round 1

Reviewer 1 Report

Comments and Suggestions for Authors

The study of Orban et al consists of a meta-analysis aimed to evaluate the efficacy of intravenous N-Acetylcysteine (NAC) in managing non-acetaminophen acute liver failure (ALF) and its impact on mortality, transplant-free survival, and length of stay in hospital. The work highlights the challenges in managing ALF and the varying efficacy of NAC across different studies and conducts a comprehensive search across multiple databases (266), identifying four randomized controlled trials meeting inclusion criteria. These four trials included patients with ALF, with various etiologies such as drug-induced liver injury (DILI), viral hepatitis, and indeterminate causes. Authors critically evaluate the findings of these four studies, but their results of meta-analysis could not demonstrate any significant benefits on overall and transplant-free patient survival in non-acetaminophen ALF, nor a significant decrease in hospital length of stay.

In their conclusions, the review underscores the need for further research to elucidate the optimal use of NAC in ALF management, clarifying dose-response relationships, identifying patient subgroups that may benefit most from NAC therapy, and standardizing outcome measures to facilitate meaningful comparisons across studies.

Comments:

-It seems to me that these four particular studies are too few to carry out such a meta-analysis, because although even two studies are a sufficient number to perform a meta-analysis, provided that those two studies can be meaningfully pooled and provided their results are sufficiently similar, the four studies used in the present work seems too dissimilar. Do the authors believe that NAC has minimal effects (as the title says) on non-acetaminophen ALF or there is an issue on different factors, such as different dosing regimens, different patients age, and specific etiologies of ALF, to draw any conclusion.

-In consequence, I believe the title of the work is not reflecting the results of the study. We do not know if NAC has minimal effects. We only know that, given our data, we cannot demonstrate any significant benefits. Authors should rephrase the title to clearly give the message: From the studies performed until now, it is not possible to conclude any beneficial effect of the treatment.

-In the Abstract section, I believe the sentence in line 16 should be: …N-Acetylcysteine (NAC) on mortality and hospital length of stay in patients with non- acetaminophen acute liver failure.

-The resolution of all tables in figures is very low. Please change resolution.

Author Response

Dear reviewer,

Thank you very much for both your time used in reading and reviewing our manuscript, as well as your input and suggestions made for the improvement of our paper which contributed a lot to the overall quality of our review. We hereby attach a point-by-point response to your comments and suggestions. All the changes made to the manuscript have been highlighted in yellow.

            Q: It seems to me that these four particular studies are too few to carry out such a meta-analysis, because although even two studies are a sufficient number to perform a meta-analysis, provided that those two studies can be meaningfully pooled and provided their results are sufficiently similar, the four studies used in the present work seems too dissimilar. Do the authors believe that NAC has minimal effects (as the title says) on non-acetaminophen ALF or there is an issue on different factors, such as different dosing regimens, different patients age, and specific etiologies of ALF, to draw any conclusion.

            A: Thank you very much for your observation as it was the main issue that led us to perform this meta-analysis. The are many case reports and case series on the benefits of NAC in non-APAP ALF but most of these studies have major drawbacks: (1) they report NAC use in ALF of different etiologies but looking at the inclusion criteria, one can see that ALF is not ALF (sometimes is just defined as an increase in serum transaminases etc.), (2) the dose and route of administration varies widely, (3) one size does not fit all, most of the major studies include different etiologies of ALF (ranging from DILI to viral, autoimmune etc), (4) mortality outcomes differ in terms of duration (ranging from 28 days to 1 years) and, moreover, when reporting “long term mortality” it may be attributed to different factors not to ALF itself (e.g. septic shock secondary to immunosuppression etc), (5) the standard of care that is used as a comparator varies widely and has changed significantly in the last two decades (from the first published trial to current days) and this may have a significant impact on patient outcome. However, there is a “standard of care” in many centers to administer intravenous NAC in patients with ALF. Current high-quality literature is composed of the 4 major studies included in our meta-analysis and, to our knowledge no other major trial will be published in the near future. Our study highlights that, to date, there is no sufficient evidence to support the routine use of NAC in non-APAP ALF and future research is needed. To conclude, our meta-analysis highlights not the benefits of iv NAC in non-APAP NAC but the need of further research in order to establish potential indications and doses (if any) that would improve survival in specific patient subpopulations. All of these have been included in our discussion section.   

Q: In consequence, I believe the title of the work is not reflecting the results of the study. We do not know if NAC has minimal effects. We only know that, given our data, we cannot demonstrate any significant benefits. Authors should rephrase the title to clearly give the message: From the studies performed until now, it is not possible to conclude any beneficial effect of the treatment.

            A: The reviewer is right; we have changed the title of our article as well as rephrased the conclusion to better reflect our results based on current data.

Q: In the Abstract section, I believe the sentence in line 16 should be: …N-Acetylcysteine (NAC) on mortality and hospital length of stay in patients with non- acetaminophen acute liver failure. 

A: Thank you for your correction. We have changed accordingly.

Q: The resolution of all tables in figures is very low. Please change resolution. 

A: We have changed the resolution of our figures and tables for a higher quality.  

Reviewer 2 Report

Comments and Suggestions for Authors

The authors have reviewed the clinical trials conducting NAC intervention to non-acetaminophen  Drug-Induced  Liver  Injury patients. 

The authors need to add PICO frame to facilitate the understanding of inclusion and exclusion criteria. 

A recent review has been published by Shirley Xue Jiang et al 2022, discussing the same topic of the submitted manuscript. In the published review, several clinical trails have been included, however, the submitted manuscript included only four studies. Is it possible for the authors to provide us with a supplementary form explaining the reason for excluding those trials from the meta-analysis? 

Author Response

Dear reviewer,

Thank you very much for both your time used in reading and reviewing our manuscript, as well as your input and suggestions made for the improvement of our paper which contributed a lot to the overall quality of our review. We hereby attach a point-by-point response to your comments and suggestions. All the changes made to the manuscript have been highlighted in yellow.

Q: The authors need to add PICO frame to facilitate the understanding of inclusion and exclusion criteria.

A: Thank you for your suggestion. We have included a PICO frame with the inclusion and exclusion criteria in order to make the meta-analysis design more easily understandable. It can now we found as table 1.

            Q: A recent review has been published by Shirley Xue Jiang et al 2022, discussing the same topic of the submitted manuscript. In the published review, several clinical trails have been included, however, the submitted manuscript included only four studies. Is it possible for the authors to provide us with a supplementary form explaining the reason for excluding those trials from the meta-analysis?

            A: Thank you for your observation. As mentioned in the study methodology we aimed at providing a high-quality meta-analysis including only randomized controlled trials. The study by Jiang SX et al. in 2022 represents a systematic review on the topic of our study. However, most of the studies included in this review were not randomised and hence they were excluded based on this criterion (the studies from “Clinical Studies: Case Reports and Case Series” subchapter). Jiang et al. also included a chapter on randomized controlled trial. We have included these trials as they fulfilled the inclusion criteria: Lee WM et al. 2009; Nabi T et al. 2017;

            The following studies reported by the authors in the randomized section were not included due to the following reasons:

  • The Acute Liver Failure Study Group. Singh S et al. 2013 report on the same patient cohort (same dataset) as Lee WM et al 2009;
  • Stravitz RT et al 2013 report on the same patient cohort (same dataset) as Lee WM et al 2009;
  • Siu JT et al 2020 is a meta-analysis of 2 RCTs

To conclude, in their systematic review, Jiang et al. reported only on two RCTs that were included in our meta-analysis. Supplementary, out database search found two additional RCTs.

Round 2

Reviewer 1 Report

Comments and Suggestions for Authors

The authors have convincingly explained all my concerns and have appropriately modified the manuscript.

Reviewer 2 Report

Comments and Suggestions for Authors

The authors have addressed the reviewer's comments and modified the manuscript accordingly.